# A deep dive into Brazilian health technology assessment: Structure, policies, and processes

Mohammed Alkhaldi[1,2,3,4,5]*, Márcia Matos[1,2,3,6], Ali Sweid[7], Aisha Al Basuoni[8], Rima Kachach[7], Maya Hassan[7], Patience Mushamiri-Kuzviwanza[9], Line Enjalbert[1,2,3], Sara Ahmed[1,2,3]

1 Faculty of Medicine and Health Sciences, School of Physical and Occupational Therapy, McGill University, Montreal, Canada, 2 Centre for Outcomes Research and Evaluation (CORE), McGill University Health Center, Montreal, Canada, 3 Centre for Interdisciplinary Research in Rehabilitation of Greater Montreal (CRIR), The Integrated University Health and Social Services Centre of West-Central Montreal (CIUSSS West-Central Montreal), Center for Outcomes Research & Evaluation, Clinical Epidemiology, Montreal, Canada, 4 Department of Public Health, School of Health Sciences and Psychology, Canadian University Dubai, Dubai, United Arab Emirates, 5 Centre for Tropical Medicine and Global Health, Nuffield Department of Medicine, University of Oxford, Oxford, United Kingdom, 6 Department of Medicine, Federal University of Sergipe, Aracaju, Brazil, 7 Faculty of Health Sciences, The American University of Beirut, Beirut, Lebanon, 8 Projects Unit, Gaza Community Mental Health Programme, Gaza, Palestine, 9 School of Public Health, University of the Witwatersrand, Johannesburg, South Africa

* mohammed.alkhaldi@mcgill.ca

## Abstract

Healthcare systems worldwide face mounting pressures from aging populations, costly medical technologies, and rising healthcare expenditures. Health Technology Assessment (HTA) has emerged as a critical tool for improving efficiency and supporting evidence-informed resource allocation through systematic evaluation. In Brazil, HTA plays a central role in advancing Universal Health Coverage (UHC), particularly through the National Committee for Health Technology Incorporation (Comissão Nacional de Incorporação de Tecnologias no Sistema Único de Saúde – CONITEC). As HTA continues to evolve in Brazil, there is an increasing need for health policy and systems research to better understand its structure, challenges, and opportunities. This study aimed to comprehensively analyze the key pillars of Brazil's national HTA system, identify existing barriers, and propose strategies to strengthen HTA processes. A mixed-methods approach was employed between 2021 and 2023, targeting HTA-related organizations and experts across multiple health sectors. Data were collected through thirteen electronic institutional surveys assessing technical aspects of HTA and nine virtual in-depth interviews exploring HTA from a policy perspective. Findings indicate a strong presence of public-sector and academic institutions within Brazil's HTA landscape, alongside broad recognition of HTA's value and CONITEC's central role in coordinating evidence generation and appraisal. However, challenges such as potential conflicts of interest and reliance on exclusive government funding were identified, underscoring the need for more diversified and sustainable financing mechanisms. The system benefits from a multidisciplinary

**Data availability statement:** The data contains identifying and potentially sensitive information related to the participating organizations and experts. Data sharing is subject to restrictions imposed by the McGill Research Board Ethics Office in Canada. Access to the data may be requested from the corresponding author or the hosting institution upon reasonable request. Requests for data access can be directed to: The Faculty of Medicine and Health Sciences Research Ethics Board (McGill IRB), McGill University: Georgia Kalavritinos, Ethics Review Administrator, REB 1, 2, 3, georgia.kalavritinos@mcgill.ca.

**Funding:** This study was supported by McGill University, Montreal, Canada, in the form of an integrated grant (MITACS grant number: FR87559, and CIHR-IRSC grant number:0478009918) and by the Lindsay Foundation Grant at CIUSSS Centre-Sud-de-l'Ile-de-Montréal in the form of a grant awarded (CIUSS grant 2021). Both grants were awarded to Dr. Mohammed Alkhaldi (grant recipient), the corresponding author of this study, whose specific roles are articulated in the "Author Contributions" section. The funders had no role in study design, data collection and analysis, decision to publish, or preparation of the manuscript.

**Competing interests:** The authors have declared that no competing interests exist.

workforce and active community participation, and HTA evidence is widely used in policymaking, particularly in evaluating clinical effectiveness, costs, and economic value. Despite these strengths, limitations persist, including insufficient institutional capacity, resource constraints, and political support. Participants emphasized the need to strengthen HTA skills, competencies, and coordination to improve the effectiveness and impact of HTA processes. This study contributes to the limited literature on Brazil's HTA system and provides evidence to inform future research and policy efforts aimed at strengthening HTA integration in support of UHC.

## Introduction

Healthcare systems worldwide have been substantially challenged by the progressive aging of populations, the development of new and expensive medical technologies, and increased expenditures. For instance, it is estimated that nearly 20% of total healthcare expenditure in the Economic Co-operation and Development (OECD) countries is inefficiently spent [1], mostly as a result of overtreatment, poor coordination, and ineffective administration. This waste may be higher in low- and middle-income countries. The research reveals that the projected impact of technological developments on health spending growth fluctuates significantly, from 10% to 75% of the yearly rise, with most studies reporting rates between 25% and 50% [2]. Health Technology Assessment (HTA) can inform effective and systematic solutions to decrease these inefficiencies and optimize the use of limited health system resources. HTA is defined as a systematic evaluation of properties, effects, and/or impacts of health technologies and interventions. It covers both the direct, intended consequences of technologies and interventions and their indirect, unintended consequences [3]. HTA is considered a bridge between the world of research and the world of decision-making by establishing interdisciplinary collaboration to evaluate and generate evidence and cost data, and its orientation to disseminate and communicate information to inform policy making [4]. The trajectories of HTA have focused on the assessment of drugs; however, it can also be applied to assessing public health interventions such as chronic disease prevention and treatment [5].

HTA is not a new process, as it was first conceptualized as early as 1976 [6]. However, there is growing interest in HTA in general and it is recommended by the World Health Organization as an essential evidence-informed policymaking approach to achieve Universal Health Coverage (UHC) and Sustainable Development Goals (SDGs) [7]. Therefore, there was an emphasis on the importance of providing technical support to countries to strengthen their HTA capacities [8]. In addition to reducing inefficiencies and waste, and contributing to the delivery of quality health services, HTA would also help in the convergence of innovation, technology, and policy, if integrated within a country's health innovation system [9]. Not only would implementing HTA help in improving the clinical decision-making process, but it is also important to enhance health surveillance and health education and support behavioral changes regarding long-term disease management. The evidence-based insights derived from

HTA are used by policymakers, healthcare providers, and patients to improve health outcomes and optimize the use of resources [10]. For this reason, adopting and implementing an HTA and making it a pillar of the health system is essential for all countries, particularly in Low- and Middle-Income Countries (LMICs) where Brazil belongs.

HTA integration in health systems is increasing not only in high-income countries but also in LMICs, which face tough decisions in terms of resource allocation and prioritizing healthcare needs due to the scarcity of resources, in addition to the growing disease burden [11]. Despite the huge need to integrate HTA into LMICs, there are many challenges to its implementation, such as the lack of data, weak local institutions, and limited technical expertise [11]. Among middle-income countries, Brazil is the largest country in Latin America, with a gross domestic product (GDP) estimated at around 7,972.5 USD per capita in 2021 [12]. Brazil has made continuous progress toward achieving universal health coverage, as the population is guaranteed the right to universal care through a decentralized public health system (Sistema Único de Saúde, SUS [13]. This system is predominantly financed through general tax revenues and contributions from federal, state, and municipal governments [14]. However, the underinvestment in the Brazilian health system has created a paradox between the citizens' right to have free health care at the primary, secondary, and tertiary levels and the increasing demands for new technologies with the underfinanced health system. This paradox has resulted in the establishment of a new framework for HTA called the National Committee for Incorporation of Technologies (Comissão Nacional de Incorporação de Tecnologias no Sistema Único de Saúde) (CONITEC) in 2011 under Law 8080 [13]. The law established regulations addressing the effectiveness and safety of new technologies, emphasized the role of clinical guidelines in their integration, and centralized HTA within the Ministry of Health (MoH), with advisory support from CONITEC [13].

According to a survey conducted in Brazil in 2019, to investigate experts' expectations for HTA development and to strengthen the impact of HTA on healthcare decisions, many challenges were reported as equipment, human resources, and financial constraints [15]. Although the study revealed many challenges, there are still considerable unrecognized gaps in HTA understanding, utilization, and practice due to the low response rate and the unrepresentativeness of the survey [15]. Brazil is a pioneer country in the field of HTA due to the presence of an institutionalized HTA framework in the country [13]; nevertheless, the evolving process of this institutionalization is not yet fully assessed and understood due to insufficient recent knowledge and literature. The motivation behind conducting this important national system analysis stems from addressing the insufficient literature about HTA and introducing novel HTA knowledge in Brazil. This study advances the existing limited evidence on HTA architecture, policy, and practice in Brazil, and it generates optimal knowledge translation strategies for useful HTA advancement at the local and federal levels. Moreover, the study findings develop reliable evidence and insights to advance and enrich the current knowledge on HTA to innovate further robust HTA systems, effective policies, and best practices in Brazil, and transfer good lessons learned to similar contexts in LMICs.

## Research aims and objectives

The overall aim of this study was to comprehensively understand the main pillars/domains of the HTA system, detailed in the objectives below, in Brazil and evaluate the current health technologies, services, and processes in the health system through the following specific objectives:

- Assess the understanding level of the HTA concept, importance, and practices among stakeholders;

- Explore the HTA stewardship and governance, capacities, resources, implementation and utilization in the health decision and policymaking process;

- Evaluate the extent to which and how current health technologies (e.g., digital devices and application, vaccines, medicines, medical equipment and supplies), and healthcare services and clinical interventions are examined using HTA;

- Identify gaps and propose feasible solutions, such as a framework to support best practices for HTA and knowledge translation strategies in national and regional arenas.

## Methods

### Ethics statement

The ethical approval from the McGill Ethics Institutional Board in Canada under the following classification was obtained in 2021. This McGill ethical approval was renewed and obtainedin June 2025. The International Ethical Standards for Biomedical Research Involving Human Persons were followed for the implementation of this work [16,17]. Two consents were obtained: 1) institutional administrative consent (electronic via email) provided by heads/leaders of the participating organizations by accepting the invitation sent via email to participate in the survey, and 2) both individual consents, written via email and verbal via phone, were obtained where participants accepted the interview invitation sent via email/phone. Additionally, participants were asked to give their second verbal consent before the interview began. Multiple consents were obtained to comply with ethical guidelines, ensure their voluntary participation, confidentiality, and the right to with-draw at any time, and their data would be discarded and not included in the final data set.

### Methodology

This national HTA system analysis was conducted between October 2021 to October 2023. The study used a cross-sectional design employing mixed methods research, including an electronic institutional HTA survey informed by a literature review, and a virtual In-depth Interview (IDI), as outlined in the protocol [18]. This critical and relevant literature included HTA-related surveys, manuals, guidelines, and frameworks. This survey was adopted from the global HTA survey developed by the WHO and subsequently adapted and advanced by the research team (MA, AAB, and SA), based on a literature review and expert consultations. To gather insights into the technical and operational aspects of HTA, the survey was administered to thirteen organizations representing diverse sectors engaged in HTA-related activities. The selection of these organizations was purposive, guided by predefined criteria detailed in the following sections. The survey was completed by designated operational team members, such as HTA officers and staff, on behalf of their respective organizations. For the IDI, nine high-level experts representing different sectors and engaged in HTA participated in interviews, which focused on policy-related aspects. The following section provides a structured discussion of both data collection tools.

### Electronic institutional HTA survey

The HTA survey was adopted from the WHO global HTA survey [19]. This electronic survey consisted of six domains that cover the pillars of the HTA system, and each domain has relevant items (questions). These domains include understanding of HTA, the use and application of HTA, implementation of HTA, stewardship and management, resource and capacity, and impediments and insights for strengthening HTA. Further domains and questions for this survey on HTA processes, standardization, and HTA decision-making were developed based on a review of recent and relevant literature. The survey questions were closed-ended questions to collect data that reflects the technical, operational, and practical aspects of HTA. One survey was administered and completed by a member designated by the team in each of the thirteen organizations involved in HTA. The full survey is enclosed with Supplement 1.

### Virtual in-depth interview

The IDI guide was developed in accordance with the best practices for qualitative research [20–22], ensuring a rigorous and systematic approach to data collection. Unlike surveys, which gather standardized responses, qualitative interviews enable an in-depth exploration of participants' policy perspectives, strategizing experiences, and decision-making processes. The IDI guide is enclosed in Supplement 2. The guide was designed using semi-structured interview techniques, providing a balance between consistency across interviews and flexibility to adapt to emergent themes. It aligns with established qualitative methodologies [23], and questions were guided by the set objectives and literature review on HTA.

Additionally, the guide adheres to recognized analysis and results reporting standards, such as COREQ (Consolidated Criteria for Reporting Qualitative Research) (S1 Checklist) and SRQR (Standards for Reporting Qualitative Research), enhancing transparency, credibility, and analytical depth in HTA research [24].

Both tools underwent a rigorous review and consultation process by ten recognized local and international experts in public health, health systems, digital technology, health economics, epidemiology, clinical specialties, and health policy and management. Feedback from these experts was incorporated into the final versions of the tools.

## Study population and sampling strategy

Maximum variation purposive sampling [25] was used for this study. The study identified two distinct groups (lists) of participants based on certain criteria. This first group consisted of twenty HTA-associated organizations, governmental, academic, private, or non-governmental, that operate within the health sector in Brazil. The second group comprised twelve experts/leaders from these organizations who are responsible for overseeing HTA policies and strategic issues, and who were selected to participate in nine individual IDIs.

Two methods were used to explore and identify names and information of main active organizations and individuals (experts/leaders): 1) a rapid review of grey and published literature, and 2) extensive consultations among research team members and with collaborators in Brazil. These methods helped to generate comprehensive lists of twenty organizations for the survey and twelve experts for the IDIs based on predefined criteria and conditions. Our target was to select up to ten existing local and national organizations and up to ten experts from the prepared list. After applying the inclusion criteria, the twenty eligible, major, active, and relevant HTA-associated organizations were identified. These identified organizations met the following conditions: the organization was officially recognized by local and national health authorities, active at least one year since its establishment, had a defined mission for HTA stewardship, production, education, research, and funding; had any previous or current programs, projects, or interventions that directly or indirectly related to HTA; or participated in one or more of HTA activities. These organizations play critical roles within the HTA framework, encompassing healthcare service delivery, regulatory enforcement, research, and evidence generation. Most of them function as both producers and users of HTA evidence, actively contributing to HTA policy formulation, decision-making, production, and use within the Brazilian health system. The organizations with no direct or indirect role in HTA and those who did not meet the inclusion criteria were excluded. A purposive sampling strategy, conducted in consultation with health authorities, was used to select the organizations that met the inclusion criteria. Email invitations were sent to twenty organizations, of which thirteen responded and consented to participate in the study. The management of these thirteen organizations was then contacted, and their representatives were asked to nominate/assign team members involved in technical and operational HTA processes within their organization, where this team is internally assigned to complete one survey on behalf of their respective organization. The inclusion criteria were also applied to the second group of experts, where twelve experts responsible for and working in HTA policy, strategy, and systems were also identified from the same HTA-associated organizations in Brazil. Those experts were HTA experts, heads of HTA, academics, policymakers, directors, or advisors. They had to be officially holding high-level positions in HTA or related to HTA at both local and national health systems, whether in HTA research, policy, management, or education. The experts with no direct or indirect role in HTA policy, strategy, and systems and who did not meet the inclusion criteria, were excluded. Mixed sampling strategies were applied to determine the final and most relevant key informants and experts from the prepared lists, which consisted of twelve experts. These sampling strategies included simple sampling, critical case, snowballing, convenience, and self-identified sampling [26]. All twelve experts were contacted by the research team via email and phone call, where nine experts consented and were invited to interviews. These sampling strategies were also guided by the application of the pre-defined selection criteria that led to the final selection of study participants, thirteen HTA-associated organizations and nine experts from the HTA community in Brazil.

The research team, led by the Principal Investigator (MA), applied this selection approach of organizations and experts to recruitment to ensure broad agreement and representation across sectors, organizations, and levels of research, policy,

managerial, operational, and technical. Given the limited number of HTA communities (i.e., organizations and experts working in HTA), the research team established a target sample size of five to ten organizations, with one expert or representative from each organization. However, to enhance representation and ensure adequate participation, additional consenting organizations and experts beyond the initial target were included.

## Data collection

The electronic HTA survey was designed using McGill's RedCap cloud-based clinical software. Each HTA associated organization of the thirteen completed the survey. This was executed by sending an official invitation outlining the research objectives to each organization's head for approval. Once approved, the research team distributed the survey via email to the nominated team leader of the organization to guide the technical, operational, practical, and managerial team/staff involved in HTA to complete the survey under the team leaders' guidance.

The research team gave all organizations and assigned teams a two-month duration to return the completed survey, offering close follow-up, feedback, and support to address any questions or issues related to the survey.

The IDIs were conducted via a web audio-video conferencing platform (Zoom Communications Inc., 2020), and each IDI lasted between 45 and 60 minutes. Nine IDIs were conducted with nine experts from policy and strategy levels from different sectors, disciplines, and levels. The PI (MA) and co-investigator (MM) in Brazil communicated with all selected experts and coordinated and conducted the virtual IDIs. The PI and co-investigator attended the IDIs, the PI led the IDI and discussion, and the co-investigator facilitated and reported the IDI.

## Data management, analysis, and sample size

All data on organization names, experts, and data from the survey and IDI were stored on a secure McGill server and only accessible by authorized members of the research team. Survey and IDI data was stored in a secured server and then imported to two software programs for data management and analysis. Survey data was analysed using the IBM SPSS Statistics version 29 software program. The survey data were analysed using descriptive statistics, including frequency distribution, percentages, categories, means, and standard deviation. Comparisons were made between organizations and sectors. IDI data was audio-recorded and then simultaneously translated and transcribed in English into MS Word sheets by the PI, assisted by trained co-investigators. Transcripts were imported into the software program, MAXQDA 12 (VERBI GmbH, Berlin), for qualitative data management and analysis. Transcripts were checked by the PI to ensure quality. Two coders and co-investigators constructed and validated codes in MAXQDA by classifying transcripts into IDI using a preset coding system that was derived from study objectives. To maintain consistency, a third independent reviewer resolved disagreements. Peer iterative review of themes, participant feedback, and triangulation with survey data were performed by the PI and reviewers to strengthen credibility. The methodological approach of this study was built based on similar studies [22,27–29]. The COREQ approach was followed for reporting the study results. The IDI transcripts were analyzed using thematic analysis, guided by both deductive and grounded theory approaches. The research team used a study framework, developed through expert consultation and literature review, based on six HTA system pillars to ensure systematic and rigorous interpretation.

The sample size was based on maximum variation and determined based on availability and accessibility, relevance, diversity, and representation of the essential organizations and experts involved in HTA, taking into account the practical, operational, technical, policy, and scientific considerations raised from the consultations mentioned above.

## Results

### Survey findings on the HTA system from a technical perspective

The thirteen Brazilian institutions completed the survey. The majority comprised public sector and academic institutions, representing 38% (n = 5) and 31% (n = 4), respectively. The private sector constituted 23% (n = 3) of the organizations

included, while one organization (8%, n = 1) belonged to the non-governmental sector. Organizational classification revealed that 28% (n = 5) operated at a national level, 6% (n = 1) in the industry sector, 28% (n = 5) were academic/non-profit entities, 22% (n = 4) were hospital organizations, and 17% (n = 3) fell into other categories. The organization's main fields of employment include Multidisciplinary (38%, n = 5), Medicine and pharmacy (31%, n = 4), Economics (15%, n = 2), and other healthcare (15%, n = 2).

All thirteen organizations demonstrated a very high understanding of the purpose and concept of HTA. Regarding the HTA application, all organizations affirmed that HTA holds a high level of importance in their respective areas of work. The majority attested to both high importance (62%, n = 8) and high application (77%, n = 10) within the organization, health system, and the specific sector of each organization (62%, n = 8). Only one organization reported weak importance and application of HTA within their organization and health system (8%, n = 1).

Regarding the central agency responsible for HTA management, twelve organizations (92%, n = 12) affirmed its existence, with the unanimous inclusion of CONITEC. The formal process for gathering HTA information to support health decision-making was reported as clearly structured and systematically managed by 85% (n = 11) of organizations.

Despite the unanimous acknowledgment of CONITEC's role in HTA production, when asked about the institutional entity where HTA is performed, the majority (92%, n = 12) identified a national committee within the MoH and a national unit or committee within the MoH. HTA reports in Brazil primarily reach the MoH (92%, n = 12), followed by a national independent committee related to HTA (23%, n = 3), clinicians' associations (23%, n = 3), patients' associations (15%, n = 2), the authority of population welfare on behalf of citizens (8%, n = 1), and others (8%, n = 1), as illustrated in Fig 1.

In terms of professional human resources involved in various HTA processes, the utilization of HTA reports varied across organizations and disciplines. Notably, the consensus on population-level health interventions was absent. Fig 2 shows the involvement of professional human resources in different HTA processes, including appraisal and decisions. The figure presents the proportion of organizations reporting the participation of each professional group in these stages. Evidence collection, synthesis, modeling, and review were most diversified across types of industry and professionals.

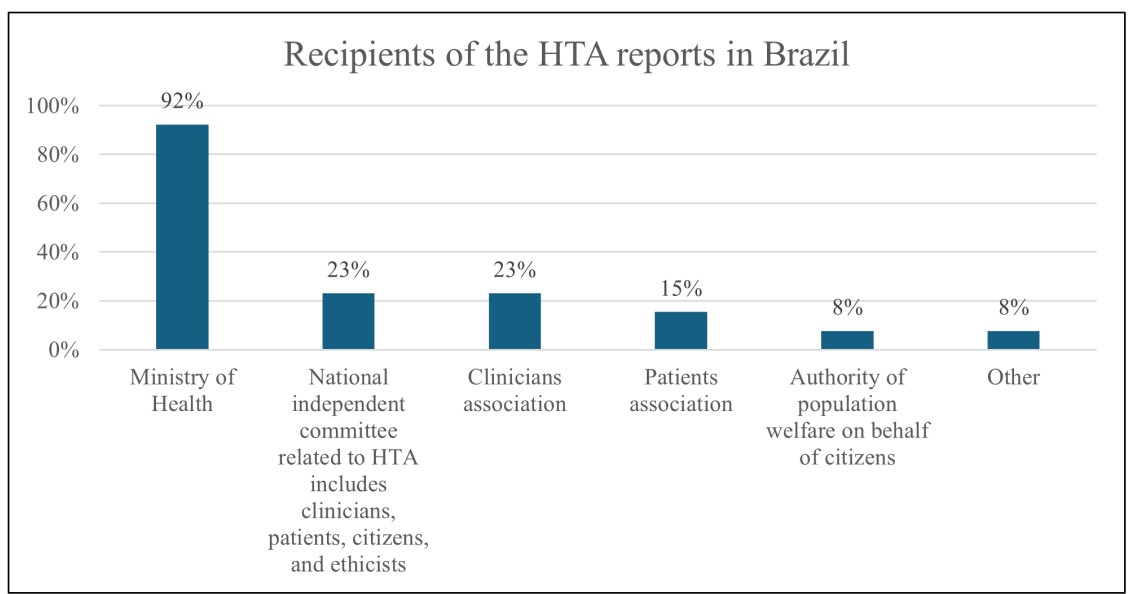

**Fig 1. Recipients of the HTA reports in Brazil (Multiple answer question).** Organizations were asked to identify institutions or groups that receive HTA reports within their national context. Reported categories reflect respondents' perceptions rather than an author-defined classification, and percentages were calculated using the total number of organizations, allowing for multiple responses.

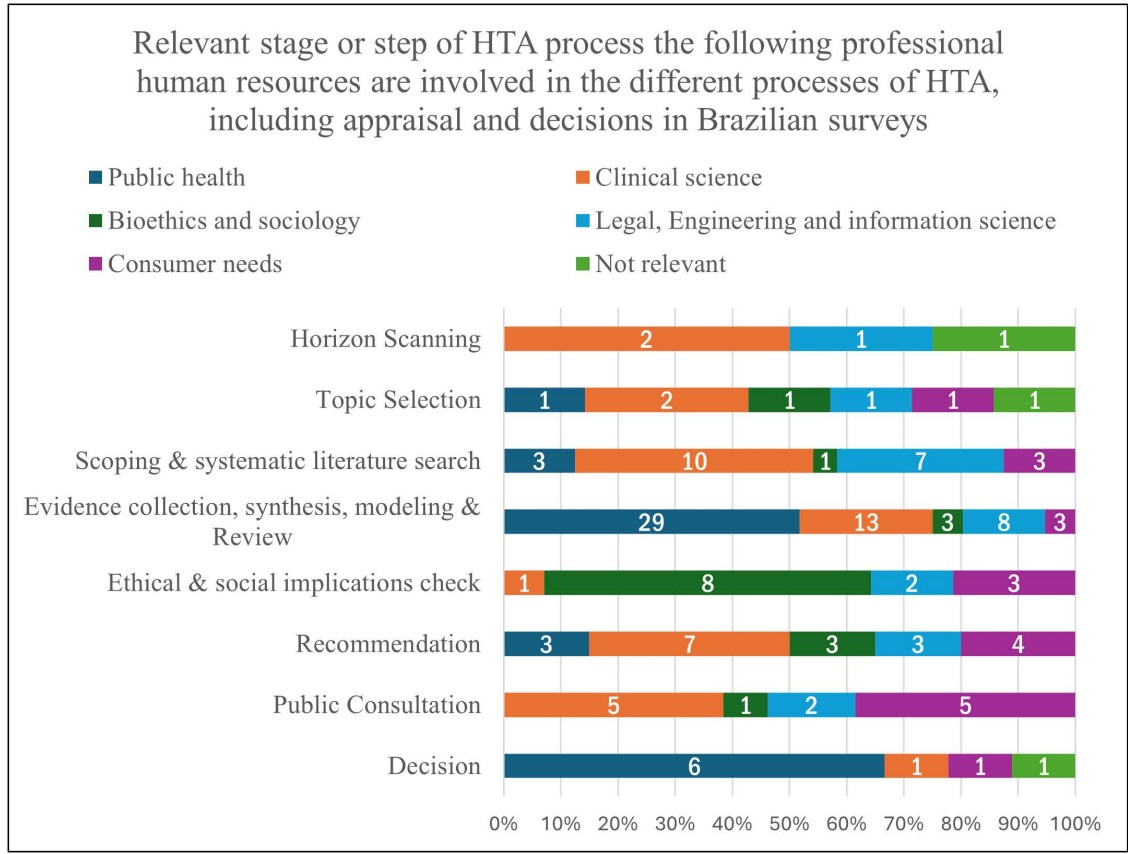

**Fig 2. Proportion of organizations indicating the involvement of professional human resources at each stage of the HTA process in Brazil, including appraisal and decision-making.**

Concerning the consideration of HTA evaluations by other organizations/countries, 92% (n = 12) acknowledged such considerations, with the UK, European Union, USA, Japan, Australia, Canada, INAHTA Members, CADTH, NICE, and IqWiG being the most cited.

All surveyed organizations (100%, n = 13) confirmed that the conclusions of HTA reports are publicly available and disseminated in Brazil. However, 75% (n = 9) acknowledged conflicts of interest declared in preparing HTA reports. Publications and dissemination occurred through public online platforms (69%, n = 9), organization websites (54%, n = 7), and gazettes (8%, n = 1). Policy outcomes based on HTA reports were reported as publicly available by 67% (n = 8) of organizations, while 33% (n = 4) answered negatively. The majority (92%, n = 11) agreed that civil society can provide feedback on HTA report recommendations. Stakeholder involvement, in reviewing draft assessment reports, 84.6% (n = 11) of organizations were acknowledged, while only 7.7% (n = 1) did not engage in such reviews and 7.7% (n = 1) did not know.

Sustainable funding allocated to HTA was affirmed by 62% (n = 8) of organizations, with 62% (n = 8) stating it is entirely government-funded. A minority reported mixed funding sources (23%, n = 3) or private funding dominance (8%, n = 1). Fig 3 illustrates the distribution of multidisciplinary professional human resources contributing to the preparation of HTA reports. Public health and clinical sciences professionals constituted the majority of contributors. In terms of the commissioning of independent agencies, 80% (n = 10) of organizations indicated that the MoH or responsible entities engage NGOs, academics, etc., for HTA preparation.

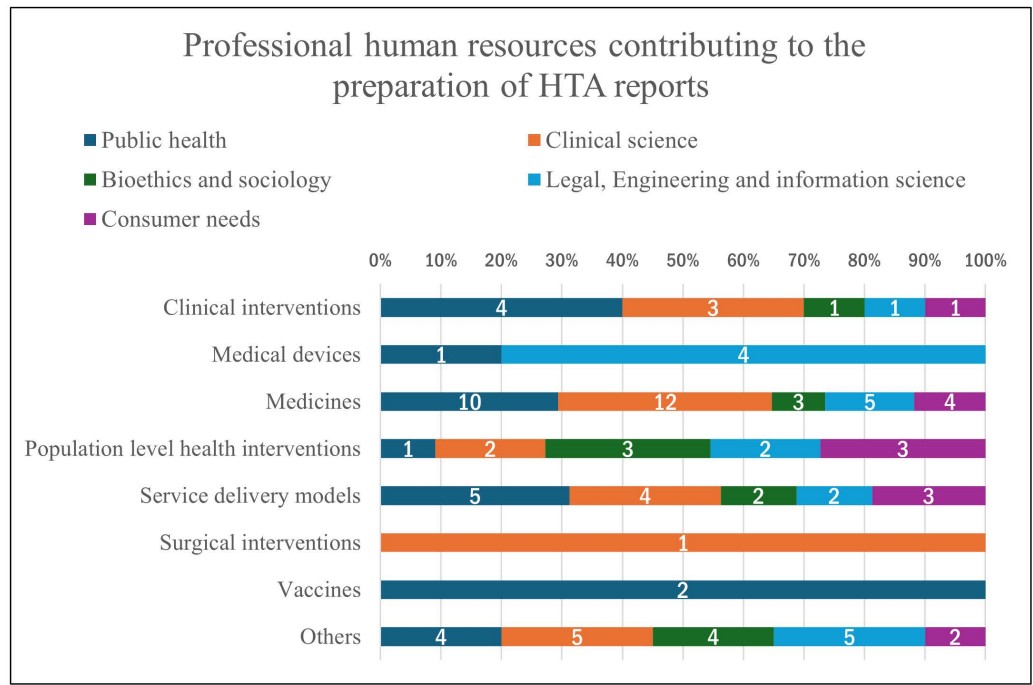

**Fig 3. Distribution of multidisciplinary professional human resources contributing to the preparation of HTA reports in Brazil, categorized by field of expertise.** (Types of professions were disaggregated into 5 categories. These categories include Public health (Epidemiologists, Biostatisticians/ Statistician, Economists/ health economists, Public Health professional), Clinical science (Medical doctor, Nurse, Pharmacist, Health professional organization), Bioethics and sociology (Sociologist, Ethicist), Legal, Engineering and information science (Biomedical and/or clinical engineer, Lawyer, Librarian/information specialist), and Consumer needs (Civil society representative, Patients representative).

All organizations unanimously agreed that clinical effectiveness, cost, and economic evaluation are the two most important dimensions of value for HTA, followed by safety (80%, n = 10), feasibility (68%, n = 9), organizational impact (56%, n = 7), equity and equality issues (48%, n = 6), environmental and political aspects (40%, n = 5), and ethical issues (24%, n = 3) (Fig 4). The majority (92%, n = 12) stated that new technology is the primary focus of assessments in Brazil, with established widespread practice, emerging technology, and further development of technology also considered. Fig 5 illustrates the aspects covered in HTA and their frequency, with clinical effectiveness being the most relevant aspect for vaccines, medical devices, and medicines.

Cost-effectiveness and economic evaluations of different technologies (medical devices, medicines, vaccines and clinical interventions) are the most important dimensions of value for HTA, which means both are well established in the HTA system in Brazil. However, equity and equality issues, ethical issues, and patient/citizen/community acceptability are less covered in HTA in Brazil, particularly for service delivery models. Organizations cited around 15 methodological guidelines published by the MoH, with international guidelines from ISPOR, EUnetHTA, INAHTA, NICE, Adophta, and RedETSA also being utilized. Guidelines indicate the availability of high-level formal and normative guidance that sets expectations and standards by topic area, while technical practices reflect the operational and methodological activities used to carry out HTA. Guidelines were reported for clinical interventions (77%, n = 10), medical devices (77%, n = 10), medicines (92%, n = 12), population-level health interventions (31%, n = 4), service delivery models (23%, n = 3), surgical interventions (31%, n = 4), and vaccines (77%, n = 10), while 8 (n = 1) % reported having no guidelines. Technical practices were reported for clinical interventions (69%, n = 9), medical devices (92%, n = 12), medicines (100%, n = 13), population-level health interventions (39%, n = 5), service delivery models (23%, n = 4), surgical interventions (46%, n = 6), and vaccines (69%, n = 9),

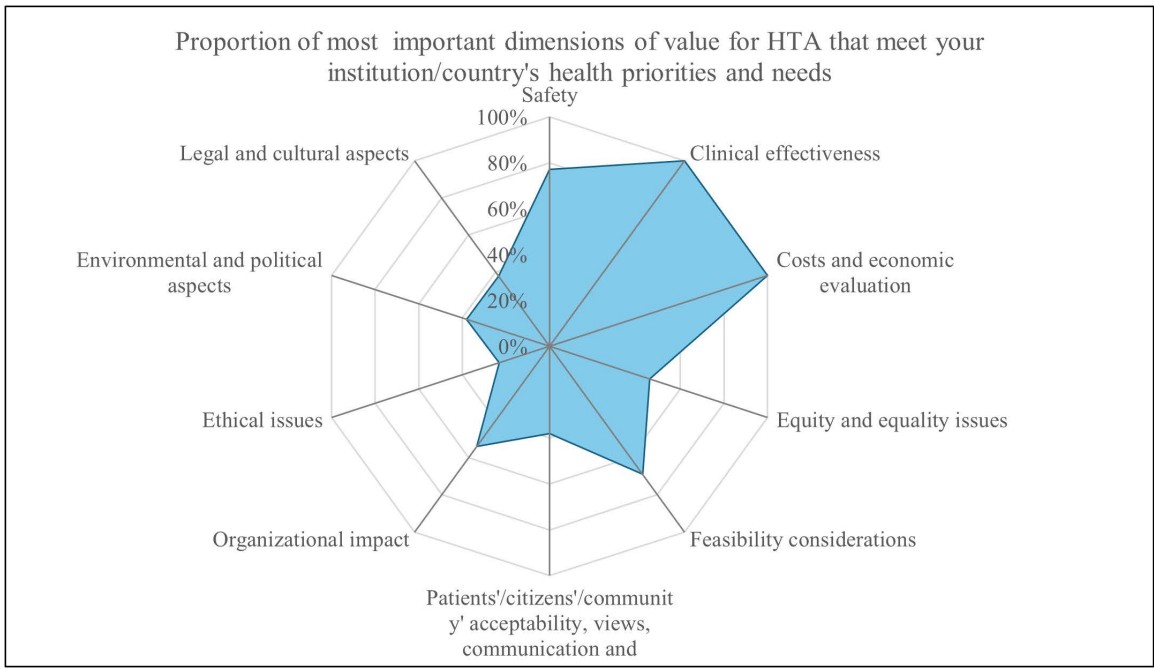

**Fig 4. Proportion of the most important dimensions of value for HTA that meet the institution/country's health priorities and needs.**

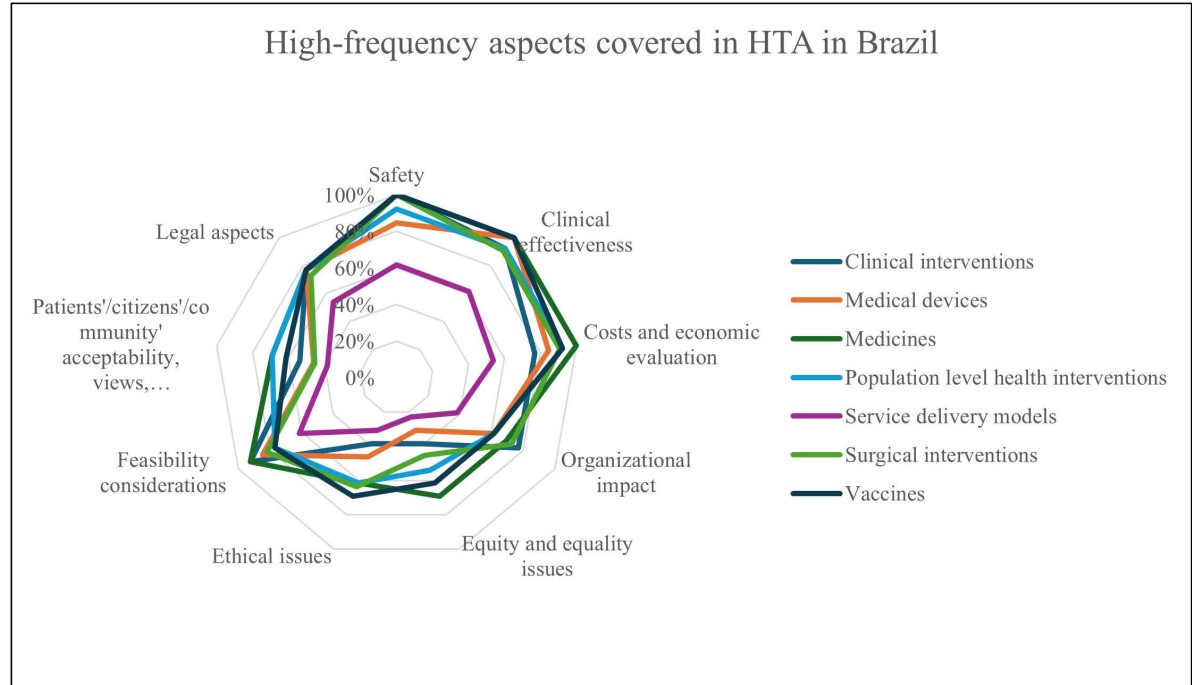

**Fig 5. High-frequency aspects covered in HTA in Brazil.**

with none reporting the absence of technical practices (Fig 6). The majority of organizations agreed there is transparency in timelines for the HTA process, as illustrated in Fig 7).

All organizations attested to information-gathering practices on new interventions to support decision-making, like HTA. Both primary and secondary data sources were considered significant by 77% (n = 10) of organizations, with 23% (n = 3) relying primarily on secondary sources. Varied methods were described for collecting HTA data, including submission by representative, authority, academia, manufacturer, and a mix of methods.

Fig 8 presents the most significant impediments at the country level to the use of HTA in health care policy decision-making, as reported by 46% (n = 6) of organizations, which were a lack of awareness and institutional capacity. Lack of political support (38%, n = 4), missing mandates from policy authorities (23%, n = 3), lack of qualified human resources (15%, n = 1), and lack of institutionalization of HTA (15%, n = 1) were also noted. To strengthen HTA capabilities and production structure, the organizations highlighted actions, such as improving qualified human resources, institutional strengthening, budget increases, and enhanced knowledge of HTA methods.

The final section of this system analysis focuses on initiatives that could strengthen HTA capacity building through academic and training programs in Brazil. Fig 9 demonstrates such needed initiatives where 92% (n = 12) reported sources/seminars/workshops, 85% (n = 11) higher education/master's degree, and 85% (n = 11) internal staff training session workshops.

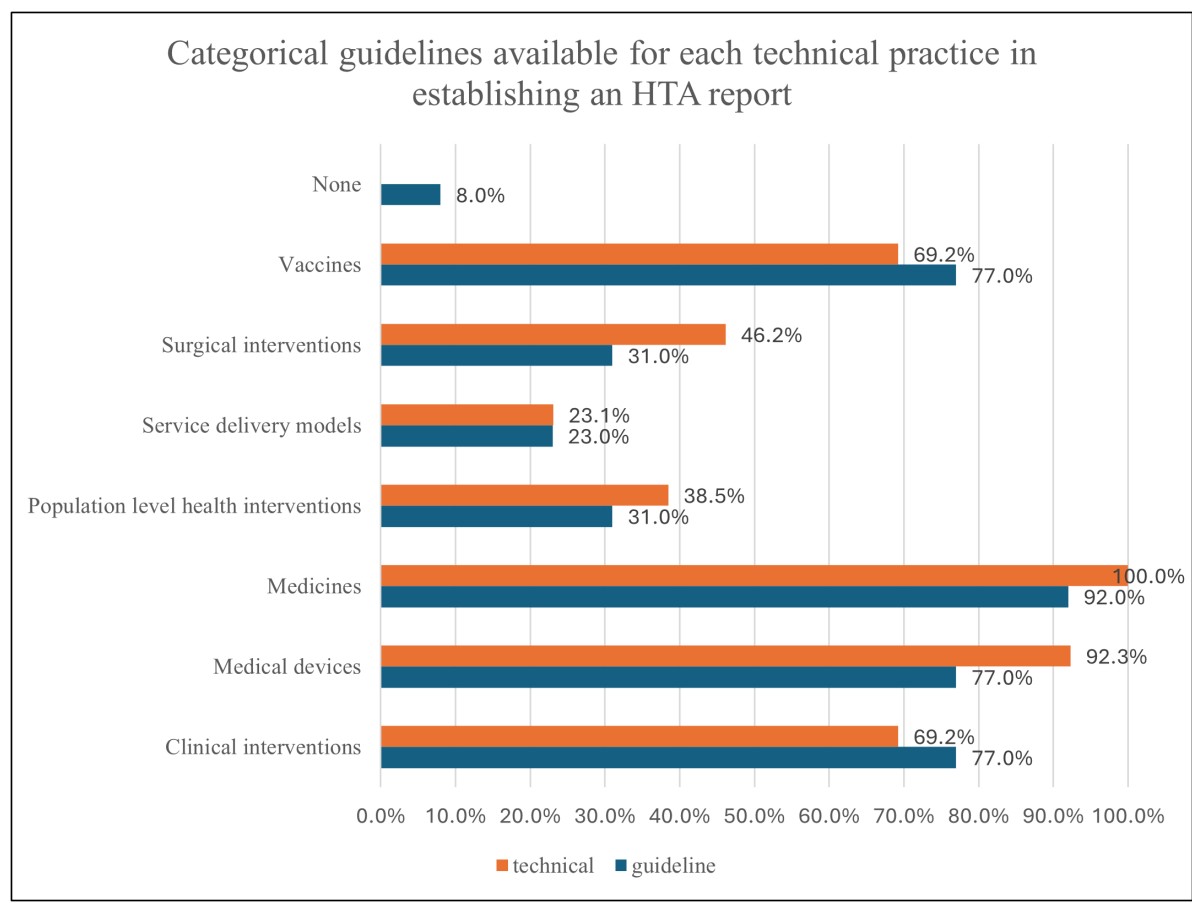

**Fig 6. Categorical guidelines are available for each technical practice in establishing an HTA report.**

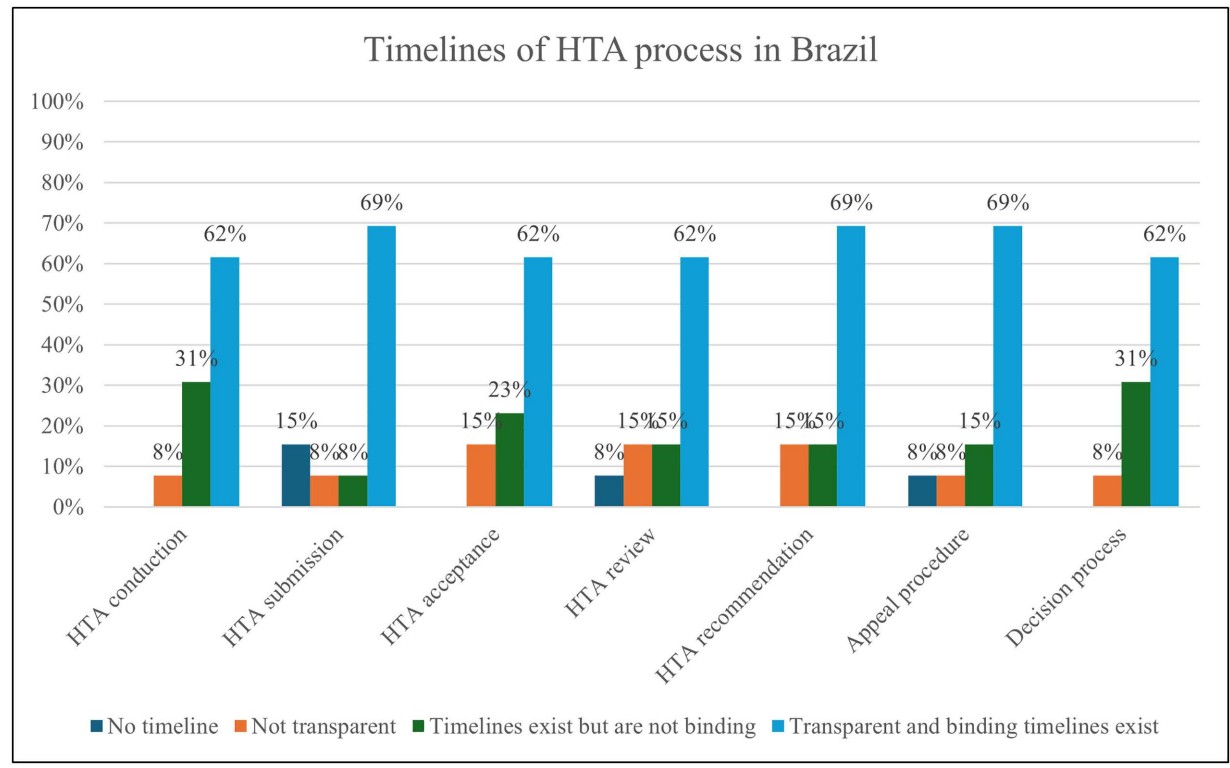

**Fig 7. Timelines of the HTA process in Brazil.**

## Qualitative evidence on HTA system: Policy perspectives

This system analysis examined qualitatively the policy perspectives of 9 experts who were interviewed: 4 academics, 3 from the public sector, and 2 from the private sector.

**Conceptualization and Understanding of HTA.** Interviews revealed diverse understandings of HTA. Definitions ranged from "evidence-based evaluation" and "evidence and decision making" to "efficiency and economic impact," "multidisciplinary approach," and "reimbursement and systematic incorporation." One academic expert described a comprehensive grasp of HTA, saying: "*HTA has been institutionalized in our healthcare system. More in the public system or we call the national health unified health system and a little bit in the private one, especially for inclusion of technologies in health care plans*" (Interviewee B3, Academic Expert).

**Perceived advantages and disadvantages of HTA.** Experts identified several advantages, including increased security and rationality in providing health technologies, decision-making support, sustainability, and a deeper understanding of new technologies. Three experts in particular noted that HTA can promote transparency in technology use, which can advance equity and improve access.

They also noted disadvantages, such as HTA overlooking alternative assessment methods, causing dissatisfaction among stakeholders who disagree with decisions, and being perceived as inefficient if not all requirements are addressed.

**Governance and stakeholder engagement.** Experts consistently identified CONITEC as the primary governance body for HTA in Brazil. The structure of CONITEC was described by one participant:"*Comissão Nacional de Incorporação de Tecnologias no Sistema Único de Saúde (CONITEC). ["Conitec's operating structure consists of two forums: the Plenary and the Executive Secretariat*"] (Interviewee B1, Public Sector).

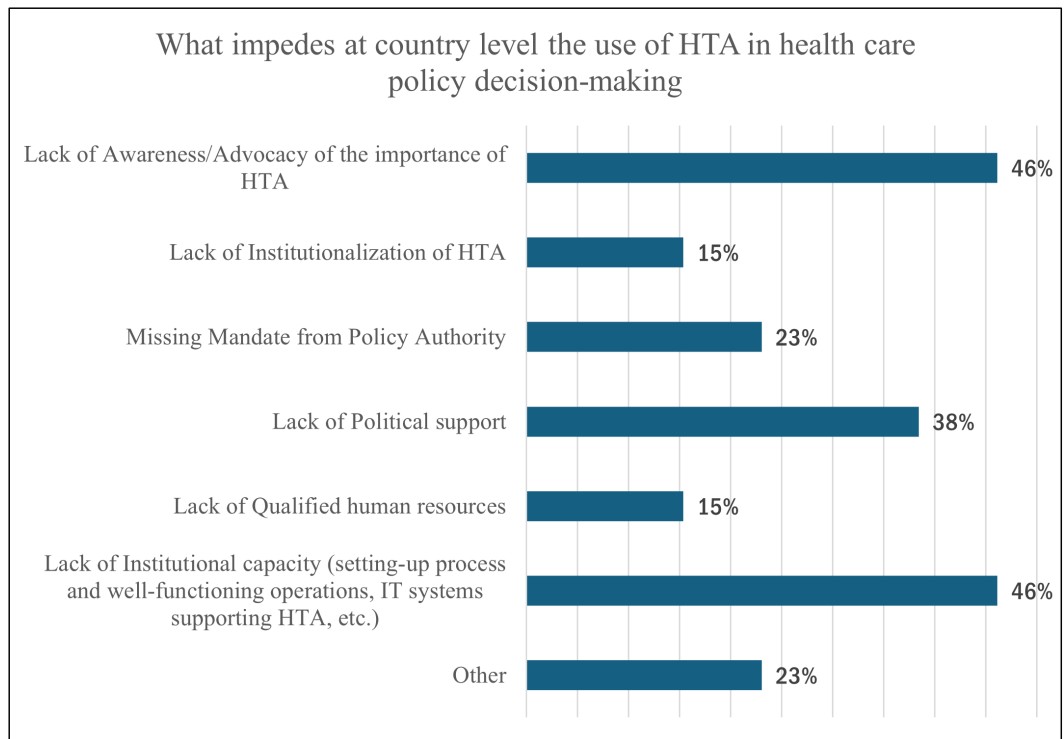

**Fig 8. Impediments of HTA use at a national level.**

Key actors within CONITEC include governmental representatives who select HTA demands, NATS (Núcleo de Avaliação de Tecnologias em Saúde), who present HTA reports, stakeholders who participate in proposal voting processes, and patients' representatives who contribute to public consultations.

**Policy framework.** Most participants indicated the existence of the national policy framework in Brazil, referencing the 2011 law and three national policies: "*The work of HTA in Brazil is related to 3 national policies: National Policy of Health Technology Management, National Policy of Pharmaceutical Assistance and National Policy of Science, Technology and Innovation in Health. They are important policies for the integration of scientific evidence into the provision of health technologies by the public health system, and to meet different health needs*" (Interviewee B1, Public Sector).

Some experts also mentioned that certain agencies bypass CONITEC due to a lack of financial resources.

**Resources and capacity.** Few participants provided detailed information regarding HTA funding. Similar to quantitative findings, governmental funding is a primary source. Challenges identified included a lack of funding sustainability and the absence of a designated budget specifically for HTA activities. In addition, limited information was available on the broader enabling environment and capacity for HTA, including institutional infrastructure, governance arrangements, human resources, and data systems that support HTA processes. One participant described CONITEC's structure and legislative framework, as well as technical staff engagement and public participation initiatives.

Participants discussed the importance of increasing representation, diversity, and involvement in HTA procedures. Talent development and the use of external consultants to enhance internal resources were also noted. One expert stated: "*CONITEC needs this external support, as many internal technicians are contracted with consultants*" (Interviewee B8, Governmental Sector).

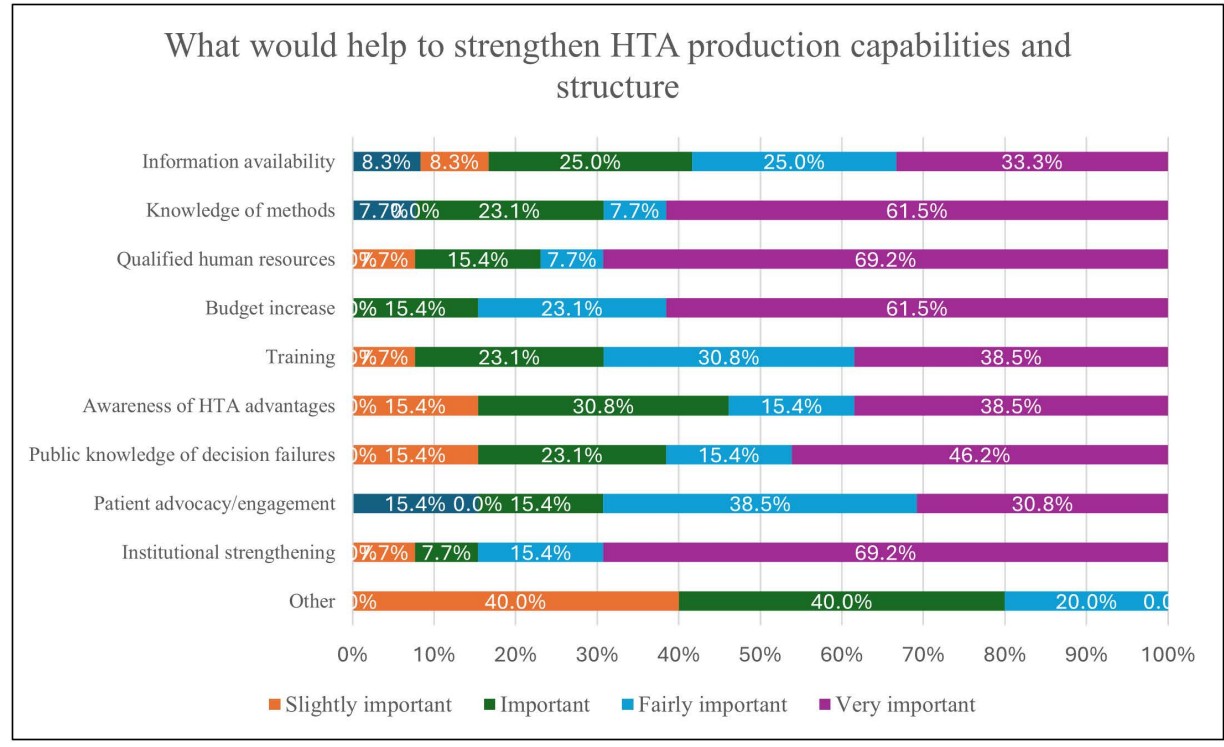

**Fig 9. Initiatives that could strengthen HTA production, capabilities, and structure.**

Participants also mentioned capacity building, workshops, the need for increased communication and public participation, specialized decision-making for rare diseases, partnerships with universities, and initiatives to strengthen the abilities and qualifications of HTA experts. One participant observed: "*The HTA process is well developed by CONITEC's technical staff, but there is still a need for greater support in the communication and qualification of society participation, which in practice happens at the initiative of patient associations and health activists*", as mentioned by the representative of the public (Interviewee B7, Public Sector).

**Implementation process.** The majority of experts indicated that the HTA process in Brazil generally aligns with international guidelines. They reported that most products, especially those that are drug-related, are subject to HTA before being offered in the public health system. Digital products and products within private health plans were also noted to undergo less rigorous assessment. As noted by a public sector representative: "*In Brazil, all products, devices, tests, and procedures are submitted to the HTA process before being offered in the public health system.*" (Interviewee B1, Public Sector).

**Influence on *decision-making*.** The influence of HTA on health decision-making was identified as a key concept by most participants. Findings highlighted that health technologies are incorporated based on clinical and economic values and safety by CONITEC. According to a government sector expert: "*In Brazil, the HTA process supports the most rational and economical offer of health technologies. Also, it offers the opportunity for society to participate in discussions on HTA. Practically all the decisions of the MoH follow the recommendations of the CONITEC to the SUS (the Brazilian Public Health System), with only one case of opposition in the latter government for the incorporation of a drug not recommended by CONITEC*" (Interviewee B1, Public Sector).

**Dissemination and communication.** The experts conveyed different levels of satisfaction with the manner of sharing and translating HTA findings in Brazil. Some expressed satisfaction with certain aspects like the availability of information and the openness of procedures, contributing to positive perceptions. A participant representing the private sector mentioned: *"Yes, I am satisfied, because all the information, such as videos of the meetings where decisions are made by committees like Cosaúde and CONITEC are public, as are the documents, sending by producer and evaluated by committees, particularly during the pandemic, and the benefit of video conferences being available on YouTube"* (Interviewee B8, Private Sector). Others showed satisfaction with the process and its impact on familiarity and qualification, as well as with information openness and the usage of HTA products for other reasons.

In contrast, some expressed concern regarding communication, integration among stakeholders, and the translation of HTA to other health areas. A participant stated: *"I think that is a major problem lack of communication, lack of integration between these different bodies, stakeholders and I think that in a less and less important part is the fact that also there are some interests in making these not well known, as well as a lack of communication and integration among stakeholders, may be driven by hidden interests"* (Interviewee B9, Public Sector).

**Capacity building and opportunities.** Most participants indicated the benefits of HTA by providing examples of its use at the institutional level, such as the development of hospital databases providing epidemiological data. At the national level, HTA was noted to bring together various stakeholders for collaborative purposes, professional training, and the development of new treatments for serious diseases. The need to expand the health budget, improve health communication, and encourage social participation in the HTA process was also mentioned. A public sector expert noted: *"Another important opportunity is the continuous process of professional capacitation"* (Interviewee B7, Public Sector).

## Discussion

The findings indicate a high level of understanding and recognition of the importance of HTA among Brazilian organizations across the public, private, and academic sectors, as reflected in stakeholders' self-reported perceptions from interviews and surveys. This underscores HTA's perceived value in healthcare decision-making. It is important to interpret these results in light of our purposive sampling strategy. The organizations included were selected specifically because of their direct or indirect involvement in HTA activities, such as evidence generation, policy advising, or technology evaluation. Consequently, the observed high level of awareness may not represent the broader health sector but rather reflects the perspectives of stakeholders actively engaged in HTA processes. The high level of understanding among relevant HTA Brazilian organizations is proven now, which should be assessed to complement it with other pillars or domains of the HTA system, as we seek to comprehensively analyze the system to understand and strengthen it. Assessment of conceptualization on HTA concept, importance, and purpose is an entry point for this comprehensive system analysis, understanding, and strengthening. Additionally, the levels of understanding of HTA remain limited and necessary to be proven to ensure the HTA is cultured, understood, and recognized among all stakeholders, policymakers, researchers, healthcare providers, suppliers, etc. Therefore, this finding remains important as it emphasizes that the sampled institutions possess not only conceptual awareness but also a consistent and comprehensive understanding of HTA's role and application across different sectors. This alignment and complementarity in the conceptual and practical aspects support the credibility of subsequent insights on implementation and system challenges.

Notably, cost-effectiveness and economic evaluation emerged as the most highly valued dimensions of HTA, followed by safety. This may be because safety is ensured through regulatory approval processes prior to the HTA assessment, prompting evaluations to focus more on comparative value and economic impact to optimize limited healthcare resources. Consistent with these findings, other aligned results confirmed that safety, clinical effectiveness, and economic evaluation are among the most frequently addressed domains within the HTA processes in Brazil [30]. CONITEC, established in 2011 by the Brazilian MoH, serves as the central, independent agency responsible for HTA management [14,20]. The widespread recognition of HTA reflects a collective commitment to evidence-based practices and resource allocation [14,31].

Brazil's mature HTA processes and the public sector's dominance in HTA mirror the country's robust, unified healthcare system since 1980 [31,32]. The Sistema Único de Saúde (SUS), Brazil's publicly funded healthcare system, via the MoH, is the main recipient of HTA reports in Brazil, followed by clinicians, patients, and citizen organizations.

CONITEC's presence ensures structured HTA governance and systematic, transparent use in decision-making. Transparency strategies are allied to social/public participation in the incorporation of health technologies through surveys, participation of the public sections on the website, applications, institutional social media accounts, and biweekly video conferences and events aimed at patients and their representatives [33,34]. However, perspectives differ on the binding nature of local vs. international evidence, highlighting the complex challenges of integrating diverse sources of evidence into decision-making frameworks. Local Data can provide more contextualized settings of the current situation, while international data brings broader evidence on the efficacy and implementation of frameworks. Although high-income countries have traditionally led the development of HTA systems, low- and lower-middle-income countries are increasingly adopting HTA processes, recognizing the value of using both local data to ensure relevance to their specific health system contexts and international data to leverage global evidence and best practices in healthcare decision-making [35,36].

The reported frequency of HTA assessments conducted annually, coupled with sustainable funding allocated to HTA activities, indicates a robust infrastructure supporting HTA in Brazil [20]. The majority of respondents agreed that HTA benefits from sustainable funding, which they defined primarily as stable public financing allocated to CONITEC through government budgets. Most indicated that HTA activities are entirely or mainly government-funded, while one organization reported that its HTA activities rely predominantly on private funding. However, this perception of sustainability largely reflects funding adequacy for CONITEC and does not necessarily indicate sufficient or diversified funding for all organizations conducting HTA in Brazil. The majority of governmental funding comes from the Healthcare system in Brazil, which is provided through a decentralized, universal SUS, financed by tax revenues and contributions from federal, state, and municipal governments [7,12]. However, the limited measurement of HTA's impact on decisions suggests a potential area for further evaluation and improvement in assessing the effectiveness of HTA processes [15].

HTA balances clinical effectiveness and economic evaluation, focusing on new technologies to keep pace with innovation. In a recent Oncology study, especially rare cancers, represented the largest proportion of evaluated demands, with the majority of technologies being adopted despite significant budgetary impacts and cost-effectiveness challenges [32]. Notably, two technologies surpassed the cost-effectiveness threshold established by the National Committee for the Incorporation of Health Technologies, illustrating the tension between innovation and affordability. Equity, ethical, and patient-centric aspects are infrequently covered, despite recent improvements [33].

Brazil has strong healthcare records, a growing number of linkage centers, and an open, modern attitude toward the use of data for research, but still faces data gaps and lacks linked, cross-validated secondary data [31,37]. Additionally, CONITEC does not use prioritization criteria to select topics for review and, despite its independence, is composed of representatives from the MoH secretariats, regulatory agencies, and other entities [14,21,32]. These institutional characteristics have been highlighted in the literature as contributing to persistent challenges related to the judicialization of health, whereby individuals seek access to health technologies not recommended by CONITEC through right-to-health litigation. While all organizations affirmed HTA reports are publicly available and disseminated in Brazil, 75% reported conflicts of interest declared, often linked to pharma and MoH requests for new assessments rather than reassessments.

HTA serves multiple roles, from clinical guidance to pricing, highlighting its multifaceted impact on various aspects of healthcare decision-making. The convergence approach adopted in initiating HTA processes underscores a collaborative model involving stakeholders at different levels, fostering inclusivity and consensus-building in decision-making frameworks [33]. Despite this, identified impediments such as lack of awareness, institutional capacity, infrastructural limitations, and insufficient political support continue to hinder the effective utilization of HTA in healthcare policy decision-making.

The presence of academic and training programs, including seminars, workshops, and higher education, demonstrated ongoing efforts to develop a skilled HTA workforce. Key areas such as cost-effectiveness analysis, HTA scope, methodology, and legal frameworks have been developed [15].

The findings elucidate Brazil's HTA landscape, showing strong institutional understanding and commitment, but also ongoing challenges in resource allocation, evidence integration, and capacity building. Addressing these challenges while leveraging existing strengths is essential for making HTA a foundation of evidence-based healthcare policy.

While this mixed-method study provides a useful benchmark for understanding HTA in Brazil, the expanded sampling that includes more provinces in Brazil is recommended to draw federal policy recommendations. Larger, more representative studies, including more institutions from each health sector, are needed for a comprehensive view of HTA in Brazil. Given the country's size and the maturity of its HTA process, the limited number of organizations interviewed is a key limitation. This sample may not capture the full range of HTA experiences nationwide, and response proportions may not be generalizable to all of Brazil. Additionally, focusing on a central HTA institution may not reflect or benefit the entire system and population equally in a country as large as Brazil.

To strengthen Brazil's national HTA system, several strategic priorities must be addressed:

• **Build Capacity and Develop Talent:**

Expand training, workshops, and partnerships with universities and external experts to boost the skills and qualifications of HTA technical staff and experts, ensuring robust institutional capacity.

• **Improve Communication, Transparency, and Governance:**

Increase transparency and accessibility of HTA findings, address gaps in coordination among HTA bodies, maintain CONITEC's independence, and clarify prioritization criteria to ensure systematic, evidence-based decision-making.

• **Enhance Stakeholder Representation and Engagement**:

Broaden participation across civil society, patient groups, clinicians, and other stakeholders by strengthening public involvement mechanisms and improving structured communication platforms beyond patient associations and activists [15,29,33,34].

• **Sustain and Diversify Funding**

Secure continued, sustainable government funding while exploring additional funding sources such as international donors and private partnerships to support HTA growth and innovation.

• **Advance Data Infrastructure and Equity Integration**

Enhance access to validated, linked healthcare data for comprehensive analysis, systematically incorporate equity, ethical, and patient-centered considerations in HTA processes, and develop specialized assessments for emerging priorities like rare diseases and digital health.

These priorities reflect both the findings of this study and global guidance on HTA system strengthening, including WHO recommendations and lessons from HTA development in LMICs [7,15,29].

## Conclusion

The study on HTA policy and technical landscape in Brazil shows widespread recognition of its importance across sectors, supported by CONITEC, the central agency responsible for HTA management established by the Brazilian MoH. This represents a shared commitment to evidence-based approaches in the healthcare system. While sustained financing, mostly from the government, supports HTA operations, problems such as insufficient quantification of HTA's effectiveness and

systemic impediments, such as a lack of knowledge and political backing impede the effective use of HTA in policymaking. Addressing these issues through focused efforts such as budget increases and capacity building is critical to increasing HTA's effectiveness. Despite strengths like academic programs supporting HTA training, areas for improvement include infrequent coverage of equity and ethical aspects in HTA and low HTA use in certain sectors like the private industry, requiring broader inclusivity and educational campaigns. Nonetheless, resolving highlighted issues while building on current strengths is critical for promoting HTA as a basis of evidence-based decision-making in Brazilian healthcare policy.

## Supporting information

**S1 Text. Survey form.**
(PDF)

**S2 Text. Interview guide.**
(DOCX)

**S1 Checklist. COREQ checklist.**
(DOCX)

## Acknowledgments

We acknowledge the support from Rebecca Zucco and Malak Alrubaie for providing additional editing and proofreading support across all sections of the manuscript. Special thanks to Dr. Aline Silveira Silva from the University of British Columbia for her invaluable guidance and support throughout the study development and implementation. We also thank all experts who contributed to the review of the tools, and all participants who volunteered their time to participate in this study.

## Author contributions

**Conceptualization:** Mohammed Alkhaldi, Sara Ahmed.

**Data curation:** Mohammed Alkhaldi, Márcia Matos, Ali Sweid, Aisha Al Basuoni, Rima Kachach, Maya Hassan, Patience Mushamiri-Kuzviwanza, Line Enjalbert.

**Formal analysis:** Mohammed Alkhaldi, Aisha Al Basuoni, Rima Kachach, Patience Mushamiri-Kuzviwanza, Line Enjalbert.

**Funding acquisition:** Mohammed Alkhaldi, Sara Ahmed.

**Investigation:** Mohammed Alkhaldi, Aisha Al Basuoni.

**Methodology:** Mohammed Alkhaldi, Aisha Al Basuoni, Sara Ahmed.

**Project administration:** Mohammed Alkhaldi, Aisha Al Basuoni.

**Resources:** Mohammed Alkhaldi, Aisha Al Basuoni, Sara Ahmed.

**Software:** Mohammed Alkhaldi.

**Supervision:** Mohammed Alkhaldi, Sara Ahmed.

**Validation:** Mohammed Alkhaldi, Line Enjalbert, Sara Ahmed.

**Visualization:** Mohammed Alkhaldi, Rima Kachach, Line Enjalbert.

**Writing – original draft:** Márcia Matos, Ali Sweid, Aisha Al Basuoni, Rima Kachach.

**Writing – review & editing:** Mohammed Alkhaldi, Márcia Matos, Ali Sweid, Aisha Al Basuoni, Rima Kachach, Maya Hassan, Patience Mushamiri-Kuzviwanza, Line Enjalbert, Sara Ahmed.

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
