## [Decision Letter · Decision Letter 0]

7 Oct 2025

PGPH-D-25-02102

A Deep Look into Brazilian Health Technology Assessment System: Structure, Policies, and Processes

Dear Dr. Alkhaldi,

Thank you for submitting your manuscript to PLOS Global Public Health. After careful consideration, we feel that it has merit but does not fully meet PLOS Global Public Health’s publication criteria as it currently stands. Therefore, we invite you to submit a revised version of the manuscript that addresses the points raised during the review process.

We look forward to receiving your revised manuscript.

Kind regards,

Paolo Angelo Cortesi, PhD

Academic Editor

Journal Requirements:

1. Please amend your detailed online Financial Disclosure statement. This is published with the article. It must therefore be completed in full sentences and contain the exact wording you wish to be published.

a) State the initials, alongside each funding source, of each author to receive each grant. For example: “This work was supported by the National Institutes of Health (####### to AM; ###### to CJ) and the National Science Foundation (###### to AM).”

For more information, please go to our submission guidelines:

https://journals.plos.org/globalpublichealth/s/submission-guidelines#loc-financial-disclosure-statement

2. Please ensure that the funders and grant numbers match between the Financial Disclosure field and the Funding Information tab in your submission form. Note that the funders must be provided in the same order in both places as well.

3. Please update your online Competing Interests statement. If you have no competing interests to declare, please state: “The authors have declared that no competing interests exist.”

4. In the online submission form, you indicated that “Anonymized data used in the current study are available upon request from the authors.”.

a) In a public repository,

b) Within the manuscript itself, or

c) Uploaded as supplementary information.

4. Please provide separate figure files in .tif or .eps format only and ensure that all files are under our size limit of 10MB.

5. We have noticed that you have a list of Supporting Information legends in your manuscript. However, there are no corresponding files uploaded to the submission. Please upload them as separate files with the item type 'Supporting Information'.

6. “S1.HTA Questionnaire, updated, 2025.pdf”, “S2.HTA Quali, Interview Guide, updated, 2025.docx” and “S3.COREQ Checklist_Brazil.docx” are currently uploaded as an 'Other' file type, which is not viewable by reviewers. Please ensure that all files meant for review are uploaded as 'Supporting Information' and include a legend in the manuscript.

Additional Editor Comments (if provided):

Reviewers' comments:

Reviewer's Responses to Questions

**Comments to the Author**

1. Does this manuscript meet PLOS Global Public Health’s publication criteria?

Reviewer #1: Partly

Reviewer #2: Yes

2. Has the statistical analysis been performed appropriately and rigorously?

Reviewer #1: N/A

Reviewer #2: N/A

3. Have the authors made all data underlying the findings in their manuscript fully available (please refer to the Data Availability Statement at the start of the manuscript PDF file)?

Reviewer #1: Yes

Reviewer #2: Yes

4. Is the manuscript presented in an intelligible fashion and written in standard English?

Reviewer #1: No

Reviewer #2: Yes

Reviewer #1: A good effort with some important and valuable recommendations made but feel it needs to be improved in terms of explanations and language throughout. Below are some examples of where I feel either explanations and/or language could be improved upon.

Abstract

• “system analysis research for system thinking and learning” – needs explained

• HTA is entirely government-funded- reads as a result but would have thought this would have been known upfront

Intro

• World Health Report 2010 – very old reference

• “Not only would implementing HTA help in improving the clinical decision-making process, but it is also important to enhance health surveillance and health education and support behavioral changes regarding long-term disease” – how exactly does HTA support this?

Research aim

• 1st part of aim is about “the main pillars of the HTA system” – not sure what is meant here by pillars

• 2nd part of aim not well expressed “evaluate the current health technologies, services, and processes in the health system”

• Objectives seem very wide ranging

Methods

• A national HTA system analysis was conducted. This could be better expressed by type of study eg survey, interviews or just research

• The survey was expanded by reviewing recent and relevant literature. It also informed the interviews. More could be said on what that literature review involved.

• Although a protocol is referenced, more detail could be provided. And whilst detail is given in the paragraphs following, the first part of the method’s section leaves the reader wondering who developed the electronic survey etc, why and how were the 13 organisations selected. This section could be better put together.

• “A mixed-method tool used to gather data from thirteen electronic institutional surveys…” reads oddly..mixed methods research was undertaken perhaps

• Whilst inclusion criteria given for selecting the thirteen HTA associated organizations, governmental, academic, private, or non-governmental, that operate within the health sector in Brazil, it would be good to know more about their respective roles in HTA or how closely aligned they are eg whilst the sector is noted, are they in service delivery or involved in research? Are they producers or consumers of HTAs -this I think is key to readers’ understanding of the results.

Results

• “All thirteen organizations demonstrated a very high understanding of the purpose and concept of HTA”. As they are HTA-related, this seems unsurprising.

• Am unclear if the organisations’ are users or providers of HTAs which has implications on how we interpret results.

• Experts were from various health sectors (what is meant by various health sectors?)

• Surprised safety is less of a concern than cost-effectiveness? Is safety a first hurdle assessed by other bodies before HTA considered? Needs explained.

• Feel some of the results would be known upfront as part of documented processes rather than via research.

• “While medical devices, medicines, vaccines, and clinical interventions are well-established areas for HTA use, equity and equality issues, ethical issues, and patient/citizen/community acceptability”. Sentence’s meaning isn’t expressed very well as the first part refers to different types of technologies, the second part to values (which could be applied to assessments of all types of technologies). Eg, consider changing it to the cost-effectiveness of different technologies is well established but less so equity etc.

Discussion

• States that the findings show a high level of understanding and recognition of the HTA’s importance among Brazilian organizations in public, private, and academic sectors. However, am still unclear how / the extent to which these organisations relate to HTA activities eg is this a biased selection in terms of their closeness to HTA?

• “Coordinated efforts are required to improve HTA efficacy..” what is envisaged by efficacy here?

Figure 1 – unclear how to interpret this as total % > 100%. Sure its fine but reader needs help to interpret.

Figure 2 – title not well explained nor are the numbers on the figure,

Figure 3 - same goes for Fig 3 regarding the numbers eg is this per organisation?

Figure 4 – some explanation needed re: safety being less of a consideration than cost-effectiveness

Language –

• title is “A Deep Look….” . More colloquial to say “A Deep Dive…”

• “system analysis study” – doesn’t seem quite right in terminology – just say a study which you then analyse using its data.

• Overall edit needed as sometimes goes into a lot of detail perhaps unnecessarily eg describing qualitative research, lots of detail on software used etc, but at other times the detail is lacking, especially a lit review mentioned twice to inform both the survey and interviews but not discussed. And whether the 13 organisations produce or consume HTAs.

Reviewer #2: The manuscript offers a timely, policy-relevant mapping of Brazil’s HTA landscape with a clear focus on CONITEC, using a mixed-methods approach that combines an institutional survey and interviews and is reported with solid ethical transparency. Also, it delivers practical insights into transparency practices, funding arrangements, and persistent capacity gaps.

There are however a number of issues. The study’s sample is very small and purposively selected (N=13), so the findings aren’t generalizable and percentages can mislead. The analysis is purely descriptive and does not compare subgroups (e.g., public vs. private). Also, because the data are self-reported, bias (e.g. social-desirability) s likely. There are also inconsistencies in dates and references (for example, CONITEC’s establishment year) and some editorial/figure issues. Finally, the evidence stops in April 2022, yet the discussion cites later developments; these should be clearly separated from the study’s empirical findings.

In my opinion, these issues could be addressed with following points:

1) The manuscript references a protocol, but additional methodological detail is needed to help readers understand the methods.

2) To clarify the methods, include a flowchart summarizing invitations, eligibility, consent, and completions (survey and interviews). If a figure isn’t feasible, provide the same counts in the Methods text.

3) The Introduction states that CONITEC was created in 2011 under Law 8,080, whereas the Discussion says “CONITEC, established in 2012 by the Ministry of Health.” Please verify the correct year and use it consistently throughout. Also, your data collection window is 3 May 2021–22 April 2022, yet the Discussion includes developments after that period (e.g., Law 14,313 of 2022 and a 2025 article on CONITEC).

4) If a figure or table underpins your main claims, include it in the main manuscript (not just the supplement). For this paper, that means keeping in the main text: the HTA pathway schematic, a bar chart of the top impediments, a timeline of key laws/CONITEC milestones, a table showing who uses HTA outputs, and a sample-characteristics table (N, sectors).

**Do you want your identity to be public for this peer review?** For information about this choice, including consent withdrawal, please see our Privacy Policy

Reviewer #1: No

Reviewer #2: No

---

## [Decision Letter · Decision Letter 1]

4 Jan 2026

PGPH-D-25-02102R1

A Deep Dive into Brazilian Health Technology Assessment System: Structure, Policies, and Processes

Dear Dr. Alkhaldi,

Thank you for submitting your manuscript to PLOS Global Public Health. After careful consideration, we feel that it has merit but does not fully meet PLOS Global Public Health’s publication criteria as it currently stands. Therefore, we invite you to submit a revised version of the manuscript that addresses the points raised during the review process.

We look forward to receiving your revised manuscript.

Kind regards,

Paolo Angelo Cortesi, PhD

Academic Editor

Journal Requirements:

Additional Editor Comments (if provided):

Reviewers' comments:

Reviewer's Responses to Questions

**Comments to the Author**

Reviewer #2: All comments have been addressed

Reviewer #3: (No Response)

publication criteria?

Reviewer #2: Yes

Reviewer #3: Partly

3. Has the statistical analysis been performed appropriately and rigorously?

Reviewer #2: Yes

Reviewer #3: I don't know

4. Have the authors made all data underlying the findings in their manuscript fully available (please refer to the Data Availability Statement at the start of the manuscript PDF file)?

Reviewer #2: Yes

Reviewer #3: Yes

5. Is the manuscript presented in an intelligible fashion and written in standard English?

Reviewer #2: Yes

Reviewer #3: Yes

Reviewer #2: I can see very major improvements in the methods section. The authors have now addressed all my comments.

Reviewer #3: I read “A Deep Dive into Brazilian Health Technology Assessment System: Structure, Policies, and Processes” with great interest. This mixed-methods analysis of Brazil’s HTA systems points to a number of strengths and weaknesses in the Brazilian HTA ecosystem, with recommendations on ways to improve its contributions in the future. The methods are well described, but more work could be done to clearly present the quantitative findings and appropriately describe the Brazilian context. I recommend that the article be accepted with minor revision.

Abstract, line 48: The sentence “HTA is entirely government-funded that requiring diverse and 49 sustainable sources…” needs editing.

Line 83 “The research reveals…”: Would recommend rephrasing for clarity. It also appears that this is also based only on OECD data

Reference 3 cites “WHO. The world health report 2012”, but there was no World Health Report released in 2012. It is unclear what the authors are referring to. The PLOS Medicine Editors. (2012). The World Health Report 2012 That Wasn’t. PLoS Medicine, 9(9), e1001317. https://doi.org/10.1371/journal.pmed.1001317

Line 116: The World Bank reference requires reformatting and appears to list Brazil’s GDP per capita as US$7,972.5

Line 125 “The law stated that…”: This is a sentence fragment that requires editing.

Line 116: The cited source does not contain information about the structure of SUS and it’s unclear how tax revenues differs from contributions from governments.

Line 310-316: Given the small numbers, I would recommend presenting these as absolute numbers

Line 416: Rephrase “academicians”

Line 456: Unclear what “environmental capacity” is referring to

Line 513: It’s not clear to me that the authors can conclude a “high level of understanding and recognition” of HTA in Brazil. It is likely true, but the survey was not set up to evaluate a degree of understanding, instead relying on self-reported understanding. At best, it would appear that authors can refer to respondent’s self-reported levels of understanding.

Line 558: It might help to interrogate what “sustainable funding” for HTA actually is. Is this sufficient public funding for CONITEC, or enough private and public funding for all organizations that want to conduct HTA in Brazil?

Line 570: Inconsistent use or non-use of abbreviations like CONITEC, LMICs, etc.

Line 587: It might help to refer to literature on persistent challenges in judicial appeals to access health technologies not recommended by CONITEC through right-to-health litigation

Figure 1: It’s unclear if this figure refers to author’s classification of institutions that report receiving HTA reports, or respondents’ perception of who receives HTA reports. This distinction, as well as the denominator for the percentages are important to clarify.

Figures 2 and 3. It took me some time to understand these figures before realizing they are describing sample characteristics. The issue with the current figures is that they appear to portray a systematic classification of all professional contributions to HTA stages and outcomes, rather than self-reporting of survey respondents’ contributions. I would recommend reformatting these figures into a descriptive table to avoid this misunderstanding.

Figure 6. What is the difference between “technical” and “guideline”? It isn’t possible to interpret the figure until this is clarified.

References often re-order organization names in first and last name format for individuals.

**Do you want your identity to be public for this peer review?** For information about this choice, including consent withdrawal, please see our Privacy Policy

Reviewer #2: No

Reviewer #3: No

---

## [Editor Report · Decision Letter 2]

15 Jan 2026

A Deep Dive into Brazilian Health Technology Assessment System: Structure, Policies, and Processes

PGPH-D-25-02102R2

Dear Dr. Alkhaldi,

We are pleased to inform you that your manuscript 'A Deep Dive into Brazilian Health Technology Assessment System: Structure, Policies, and Processes' has been provisionally accepted for publication in PLOS Global Public Health.

Best regards,

Paolo Angelo Cortesi, PhD

Academic Editor